# Perinatal Whole Blood Zinc Status and Cytokines, Adipokines, and Other Immune Response Proteins

**DOI:** 10.3390/nu11091980

**Published:** 2019-08-22

**Authors:** Julie Nyholm Kyvsgaard, Christina Ellervik, Emilie Bundgaard Lindkvist, Christian Bressen Pipper, Flemming Pociot, Jannet Svensson, Steffen Ullitz Thorsen

**Affiliations:** 1Copenhagen Diabetes Research Center (CPH-DIRECT), Department of Paediatrics, Herlev Hospital, University of Copenhagen, Herlev Ringvej 75, 2730 Herlev, Denmark; 2Department of Production, Research, and Innovation; Region Zealand, Alleen 15, 4180 Sorø, Denmark; 3Department of Clinical Medicine, Faculty of Health and Medical Sciences, University of Copenhagen, Blegdamsvej 3B, 2200 Copenhagen N, Denmark; 4Department of Laboratory Medicine, Boston Children’s Hospital, Harvard Medical School, 300 Longwood Avenue, Boston, MA 02115, USA; 5Department of Public Health, Section of Biostatistics, University of Copenhagen, Copenhagen, Oester Farimagsgade 5, 1710 Copenhagen K, Denmark; 6Steno Diabetes Center Copenhagen, Niels Steensensvej, 2820 Gentofte, Denmark

**Keywords:** zinc, cytokines, adipokines, TREM1, C-reactive protein, mannose-binding lectin, infant, newborn, immune system

## Abstract

(1) Background: Zinc is an essential micronutrient and zinc deficiency is associated with immune dysfunction. The neonatal immune system is immature, and therefore an optimal neonatal zinc status may be important. The aim of this study was to investigate the possible association between neonatal whole blood (WB)-Zinc content and several immune markers. (2) Methods: In total, 398 healthy newborns (199 who later developed type 1 diabetes and 199 controls) from the Danish Newborn Screening Biobank had neonatal dried blood spots (NDBS) analyzed for WB-Zinc content and (i) cytokines: Interleukin (IL)-1β, IL-4, IL-6, IL-8, IL-10, IL-12 (p70), interferon gamma, tumor necrosis factor alpha, and transforming growth factor beta; (ii) adipokines: leptin and adiponectin; (iii) other immune response proteins: C-reactive protein (CRP), and mannose-binding lectin (MBL), and soluble triggering receptors expressed on myeloid cells1 (sTREM-1). WB-Zinc content was determined using laser ablation inductively coupled plasma mass spectrometry. For each analyte, the relative change in mean level was modelled by a robust log-normal model regression. (3) Results: No association was found between WB-Zinc content and all the immune response markers in either the unadjusted or adjusted models overall or when stratifying by case status. (4) Conclusions: In healthy Danish neonates, WB-Zinc content was not associated with cytokines, adipokines, CRP, MBL or sTREM, which does not indicate a strong immunological function of neonatal zinc status.

## 1. Introduction

Zinc (Zn) is an essential trace metal for a plethora of metalloenzymes involved in growth, cognition, reproduction, and immune function [1]. Zinc serves as an antioxidant through several mechanisms [2]. For examples, the zinc ion has a structural role in stabilizing copper/zinc superoxide dismutase, thereby increasing the redox potential of copper ion (Cu (II)) resulting in the breakdown of free radicals, but zinc is not redox active and cannot catalyze Fenton chemistry [3]. Further, zinc induces the cysteine-rich zinc-binding protein metallothionein, which serves as an antioxidant [4].

In Scandinavia, including in Denmark, approximately 6% of the population have inadequate zinc intake [5], which is low compared to many African or middle Eastern countries [5]. The primary reasons for zinc deficiency are inadequate dietary intake, impaired resorption [6], or increased physiologic demands (e.g., pregnant women) [5,7]. Physiological plasma zinc levels are estimated to be approximately 12–16 μM [8]. Zinc deficiency is associated with impaired growth and cognition [9], dysregulation and impairment of both the adaptive and innate immune system [6,10,11], systemic inflammation [11], and increased susceptibility to infectious diseases [6]. These harmful effects have fortunately been shown to be reversible, as zinc supplementation restores or even improves immune function [6].

In an observational study of adults, increases in serum zinc are inversely associated with C-reactive protein (CRP), interleukin 6 (IL-6), and tumor necrosis factor alpha (TNF-α) [12]. In randomized trials, zinc supplementation has been shown to decrease the incidence of infections in the elderly [13] and reduce morbidity and mortality in preterm neonates [14]. In the elderly, zinc supplementation decreases markers of oxidative stress (such as 8-oxodGuo, malondialdehyde, and 4-hydroxyalkenals) [13,15], the chronic inflammatory marker CRP [15], and the inflammatory cytokine IL-6 [15], and increases interleukin 2 (IL-2) which is important for cell-mediated immune responses [13]. In a study of Mexican children aged 6–7 years old, with plasma zinc within the reference range, a supplementation of zinc and other micronutrients increased the ex vivo monocyte generation of IL-2 and interferon gamma (IFN-γ) and decreased the generation of IL-10 compared to before supplementation [16]. In Indonesian infants aged 3–10 months old, stimulated ex vivo IL-6 production was decreased in zinc-deficient infants compared to non-deficient infants [17].

Proinflammatory cytokines (such as IFN-γ, TNF-α, and interleukin 1β (IL-1β)) promote beta-cell destruction and exacerbate autoimmunity, which may ultimately lead to type 1 diabetes (T1D) [18]. It is debatable whether zinc deficiency is associated with T1D [19,20]. However, no study has investigated the association between immune response markers and zinc status in healthy newborns who later develop T1D and newborns who do not.

The neonatal immune system is immature, and it is well known that adaptive immune responses are deficient in early life. Further, there are numerous significant deficiencies of the innate immune system in neonatal life. The consequence of this is increased susceptibility to infections, and therefore increased mortality and morbidity in early life [21]. Environmental exposures in the perinatal period, such as nutrients, are shown to induce changes in immune function [22]. As zinc deficiency compromises the immune system, it is relevant to investigate the association between zinc status in healthy/non-infected newborns and a large number of cytokines, adipokines and other immune response proteins. Therefore, this study aimed to investigate this association using neonatal whole blood zinc (WB-Zinc) content in dried blood spots from the Danish Newborn Screening Biobank (DNSB). This study may lead to better insight into the role of zinc in the perinatal immune system.

## 2. Materials and Methods 

### 2.1. Study Design and Sampling

This population study is based on a case-cohort design from the DNSB, located at Statens Serum Institut, Copenhagen, Denmark. Newborn dried blood spots (NDBSs), from children born in Denmark between January 1991 and November 1998, were taken as capillary blood samples from the newborn’s heel, 5–7 days after birth, and immediately transferred to filter paper. NDBSs offer screening of several diseases in Danish newborns. The NDBSs are stored at −20 °C/−4 °F in the DNSB. This biobank includes close to 100% of the NDBSs of Danish newborns since 1982 [23]. The DNSB is mostly comprised of Danish ethnic children. In a study (*n* = 3956), ~94% of the newborns had mothers of Danish origin, 1.4% had mothers from other western countries and the remaining newborns had mothers from non-western countries [24].

Individuals were randomly selected from an original case-control study, conducted by our research group, aiming at examining predictors of childhood and adolescent T1D [25]. The focus of the current study is the association between zinc and a large number of cytokines, adipokines and other immune response proteins in healthy/non-infected newborns. Cases were those who later developed T1D, and controls were those who did not. Briefly, T1D diagnoses were retrieved from the Danish National Patient Registry based on International Classification of Diseases (ICD )codes (ICD-8 249 and 250. ICD-10 DE 10.x–14.x) and were cross-validated using the Danish Childhood Diabetes Registry (DanDiabKids) to secure a T1D diagnosis [26]. DanDiabKids, established in 1996, is a nationwide register, collecting data from all Danish pediatric diabetes clinics treating patients with T1D aged 0–18 years.

Cases were matched with controls based on date of birth. In total, 398 participants (199 cases and 199 controls) were included in the study, though 17 controls and 1 case were excluded from the primary pre-selected analysis due to missing covariate data (Figure 1).

### 2.2. Exposure Assessment 

#### Assessment of Whole Blood Zinc Content

The NDBSs, measuring 3.2 mm in diameter and containing ~3.4 µL of dried capillary blood, from each participant, were analyzed for zinc (^66^Zn) using laser ablation inductively coupled plasma mass spectrometry (LA-ICP-MS) [19]. Additionally, the NDBSs were analyzed for potassium (^39^K) in the same run. ^39^K was used to adjust for possible differences in blood volume in the NDBSs and the hematocrit value in the newborn [27,28]. The NDBSs were analyzed by continuous line scans through the center of each sample, 12 consecutive points on each NDBS, which aimed to increase the precision of the WB-Zinc content. All of the samples were analyzed in the same analytical run, thereby minimizing operational variation. 

During the study period (1991–1998) the American filter paper 903 (Whatman BFR, St. Louis, MO, USA) were used. By only including cases and controls from this period, we removed the risk of including filter paper, by different manufacturers, with possible zinc content differences. For further details regarding analytic setup, see Kyvsgaard et al. [19].

### 2.3. Outcome Assessment

The NDBSs were analyzed using a multiplexed sandwich immunoassay based on flowmetric Luminex xMAP^®^ technology, which can measure up to 25 inflammatory markers simultaneously on NDBSs [29]. The following analytes were quantified: (i) cytokines; IL-1β, IL-4, IL-6, IL-8, IL-10, IL-12 (p70), IFN-γ, TNF-α, and transforming growth factor beta (TGF-β); (ii) adipokines; leptin and adiponectin; and (iii) other immune response proteins: CRP and mannose-binding lectin (MBL) involved in innate immunity, and soluble triggering receptor expressed on myeloid cells1 (sTREM-1) belonging to the immunoglobin superfamily.

In all assays, matched pairs were run together to avoid batch effects/interassay variation [30]. Biomarker analyses are described in detail elsewhere [29]. Quality control of the analysis was made using mouse IL-6 as an internal analyte added to the extraction buffer to detect pipetting errors, and biotinylated beads to detect signal errors (a more detailed description are found in Skogstrand [31]). Calibration curves were used on each plate together with one high and two low controls. Samples, calibrators, and controls were analyzed in duplicates.

The absolute levels of the cytokines, adipokines and other proteins involved in the immune response stratified by case status are presented elsewhere [32].

### 2.4. Other Variables 

A number of descriptive variables and possible confounders were available for inclusion in our analyses. These variables are listed and the coding of them is presented in Table 1.

### 2.5. Statistical Analysis 

The random sampling of 400 individuals (200 cases and 200 controls) from the larger cohort of 2086 cases and 4172 controls [25] was performed by a random number generator in Statistical Analysis Software (SAS) version 9.3 using the floor-statement and was performed on the cases born from 1991–1998 (as the same filter paper was used in this period). Each case was originally matched with 2 controls based on date of birth but, for this nested study, we only included the first control found for each case.

Neonatal WB-Zinc content was log2-transformed, due to a log-normal distribution. Results are therefore interpreted as relative change (RC) in mean levels of immune response markers for each doubling in neonatal WB-Zinc content. We analyzed the data overall and stratified by case status.

Neonatal WB-Zinc content was modeled by a robust log-normal model regression, taking into account: (i) that measurements are potentially both left and right censored; and (ii) correlation within immunoassay. To account for correlation within the immunoassay, an inference was based on a working independence generalized estimation equation (GEE) approach.

The simultaneous evaluation of neonatal WB-Zinc content on all analytes was performed using the model stacking approach detailed in Pipper et al. [33].

Subsequent adjustment for multiple testing and familywise 95% confidence bands (95% CI) were calculated using the single-step procedure by Hothorn et al. [34]. GEE estimates of mean ratios and accompanying confidence limits were calculated on a log-scale and transformed back to the original scale.

Model selection was performed prior to statistical analyses based on our previous work using these data: (i) univariate models; and (ii) primary adjusted models using possible confounders (covariates associated with neonatal WB-Zinc content (birth weight and birth year), see Kyvsgaard et al. [19]). Due to the construction of our study sample, we also included case status in the multivariate models. 

Overall functional misspecification by including neonatal WB-Zinc content as a trend (linear variable) was assessed by a lack-of-fit test. Specifically, we included a quadratic term of neonatal WB-Zinc content and tested its significance by a robust Wald test.

*p*-values were evaluated at the two-sided 5% significance level and we decided prior to our statistical analyses both to include results with and without adjustment for multiple testing.

All analyses were made using the statistical software package R version 3.5.1 (the R foundation for statistical programming, Vienna, Austria) and the add-on packages, survival and multcomp.

### 2.6. Ethics

The study was performed in accordance with the Helsinki II Declaration. Furthermore, the study was approved by the Danish Ethical Committee (H-2-2014-007) and by the DNSB Steering Committee. According to Danish law, anonymous studies do not require further informed consent. 

## 3. Results

Characteristics of the study population are presented in Table 1. The overall median (Q1, Q3) neonatal WB-Zinc content was 1.880 × 10^−3^ (1.142 × 10^−3^, 2.545 × 10^−3^) units.

### 3.1. Unadjusted Models and Adjusted Models

No association between neonatal WB-Zinc content and cytokines, adipokines and other proteins involved in immune response were found in the unadjusted and adjusted analyzes without correction for multiple testing (Table 2). Results remained the same after correction for multiple testing (Appendix A). Non-linear models were considered for the above-mentioned primary exposures. A quadratic term did not improve model fit, and so these exposures were retained as linear predictors.

### 3.2. Unadjusted Models with Statification on Case Status

We found no signs of effect modification by case status (neonates who later developed T1D versus neonates who did not) on our main effects. Results are presented in Table 3.

## 4. Discussion

In this study, we did not find any association between neonatal WB-Zinc content and levels of cytokines (i.e., IL-1β, IL-4, IL-6, IL-8, IL-10, IL-12 (p70), IFN-γ, TNF-α, and TGF-β), adipokines (i.e., leptin and adiponectin) and other proteins involved in the immune response (i.e., CRP, MBL, and sTREM-1) in healthy newborns. These are novel findings. 

### 4.1. Comparison with Other Studies

A randomized controlled trial included newborns with sepsis, and one group was zinc-supplemented (i.e., 3 mg/kg of zinc sulfate monohydrate twice a day orally for 10 days) on top of standard treatment with antibiotics (*n* = 134, age < 28 days). The zinc-supplemented group of neonates had significantly higher serum zinc levels and a significant decline in the concentration of the inflammatory cytokines IL-6 and TNF-α. The control group only had a significant reduction in IL-6 after treatment with antibiotics. Mortality was lower in the zinc group, though it was not statistically significant (*p* = 0.12) [35]. This study suggests that zinc has anti-inflammatory properties. The study differs from ours by including infected newborns and by using zinc supplementation as intervention. Zinc supplementation lead to a higher variation in serum zinc levels between the two groups of newborns, thereby increasing the power and limiting the risk of type 2 errors (false negative results). A murine study found that the IL-6 functions as an enhancer of the zinc importer zinc transporter protein (Zrt)-, iron-regulated transporter (Irt)-related protein (ZIP)14 expression on hepatocytes, resulting in lower plasma concentrations of zinc when inflammation was induced [36]. 

A study, which included a cohort of healthy breast-fed Indonesian infants, had their serum zinc analyzed at age 3, 6 and 9 months. Further, biomarkers for inflammation (i.e., CRP and α-1-glycoprotein) were measured. This study showed that infants with lower zinc status had higher CRP levels. This study also highlights that zinc deficiency is overestimated when the infants are in a condition with inflammation [9], which, as mentioned above, may be mediated by IL-6, etc. [36]. We could not confirm an association between levels of neonatal WB-Zinc content and CRP, perhaps due to the differences in age between our studies. At 3–9 months of age, zinc status is primarily dependent on the infant’s diet and not dependent on the mother’s nutritional status, as is the case when a fetus [37]. In addition, the cohort of infants were from Indonesia, where approximately 30% of the population have an inadequate zinc intake, compared to only 6% in Scandinavia [5]. Maternal zinc status is dependent on the nutritional status and/or chronic infection [38]. Maternal zinc deficiency can lead to a reduction in placental zinc transport to the fetus, thereby inducing zinc deficiency in the fetus. However, this is only observed when the maternal zinc deficiency is severe due to the fact that the active transport of placental zinc to the fetus is maintained even though maternal zinc levels are lowered [1]. 

A study, including 88 healthy mothers who gave birth to healthy newborns with a gestation age of 38–42, showed that maternal serum leptin levels were positively correlated with maternal zinc levels. However, serum leptin levels in cord blood were not found to be associated with serum zinc in cord blood [39]. The latter result is in line with findings from our study.

In elderly individuals, zinc supplementation decreases circulating inflammatory cytokines, CRP and other plasma oxidative stress markers [12,13,15]. A randomized, double-blind, placebo trial included healthy elderly individuals (*n* = 40, aged 56–83). The intervention group received 45 mg zinc/day for 6 months, which lead to an increase in plasma zinc, and a decrease in CRP and IL-6 levels [15]. A similar randomized, double-blind, placebo-controlled study included healthy elderly individuals (*n* = 50, aged 55–87). The intervention group received 45 mg zinc/day in total in 12 months. Zinc supplementation lead to a significantly lower incidence of infections, an increase in plasma zinc, and a decrease in TNF-α levels and oxidative stress markers [13]. Further, this study showed that older individuals had significantly lower plasma zinc and increased oxidative stress compared with a group of younger adults (*n* = 31, aged 18–54) [13]. Furthermore, a cross-sectional study included healthy Rural Korean Adults (*n* = 1,055 i.e., 404 men and 651 women, aged > 40). These subjects had their dietary zinc intake analyzed. No significant inverse relationship between dietary zinc intake and inflammatory cytokines/markers IL-6, TNF-α, and CRP were shown [12]. These studies [12,13,15] suggest that the elderly population differ from newborns due to the increased level of oxidative stress and higher risk of zinc deficiency, which can explain why studies that include elderly individuals are able to show a significant decrease in levels of inflammatory cytokines and other inflammatory markers when zinc supplemented. Further, randomized controlled trials are more powerful compared to observational study designs.

### 4.2. Strengths and Limitations

This study is larger than many of the studies previously published on the association of early life zinc exposure and immune response markers [9,14,16,17,35,39]. Further, we report findings from a population of neonates with no sign of serious infections (CRP range of 0.01–20.1 mg/L). A limitation is pre-analytical factors, which could falsely elevate levels of zinc, such as hemolysis, starvation, and diurnal variation [1]. In particular, dried blood spots are prone to hemolysis and, unfortunately, we did not have serum values. The zinc level for each participant was divided with the potassium level, thereby adjusting for hematocrit, thereby limiting hemolysis bias [28]. The study is cross-sectional without serial data on zinc and cytokines, adipokines and other proteins involved in the immune response, which could have given further information on long-term associations. Also, we did not have information on the mothers’ zinc or nutritional status, thereby we were not able to investigate the link between the mothers’ and newborns’ zinc status and immune response. A reason for why no associations were found might be due to the fact that zinc deficiency is rare in Denmark ~ 6% [5], but our intent was also to examine associations within the normal physiological span. Residual confounding cannot be excluded either. 

### 4.3. Future Perspective

No link between neonatal WB-Zinc content and cytokines, adipokines and proteins involved in the immune response was found in this study of healthy neonates. In future research, it would be interesting to conduct serial analyses of zinc status including other micronutrients during pregnancy and in the neonatal period in relation to immune response markers, due to possible complex interactions between micronutrients and the immune system. It may be relevant to include subgroups of newborns with an increased risk of zinc deficiency (i.e., premature [1,40]) or newborns with low birth weight [41] in future research. 

## 5. Conclusions

In healthy/non-infected neonates, WB-Zinc content was not correlated to cytokines, adipokines, CRP, MBL or sTREM-1.

## Figures and Tables

**Figure 1 nutrients-11-01980-f001:**
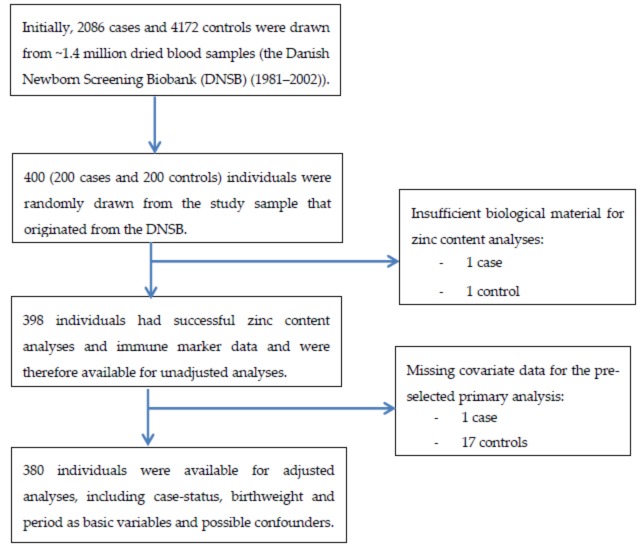
Flow chart of participants included in the study and different models.

**Table 1 nutrients-11-01980-t001:** Characteristics of the study sample (including specification of missing values). Cases are children who developed childhood type 1 diabetes (T1D), and controls who did not develop T1D during childhood.

Variables	Cases (*n* = 199)	Controls (*n* = 199)
Sex ^1^		
Female, *n*/% of total	103/51.8	95/47.7
Gestational age ^2^,		
Median/interquartile range (IQR), weeks	40.0/1.0	40.0/2.0
Birth weight ^3^,		
Median/IQR, grams	3500/630	3500/744
Maternal age ^4^,		
Median/IQR, years	29.0/6.0	28.0/8.0
Season of blood sampling, *n*/% from total		
Winter	41/20.6	40/20.1
Spring	49/24.6	49/24.6
Summer	59/29.6	60/30.2
Autumn	50/25.1	50/25.1
Period of blood sampling, *n*/% from total		
1991–1993	109/54.8	109/54.8
1994–1998	90/45.2	90/45.2
Human leukocyte antigen (HLA)-risk groups ^5^, *n*/% from total		
High/moderate ^6^	151/82.5	68/40.0
Low/protective ^7^	32/17.5	102/60.0

Missing data (*n*/% of total): ^1^ 11/2.8, ^2^ 17/4.3, ^3^ 24/6.0, ^4^ 11 17/4.3, and ^5^ 45/11.3. HLA-DQB1 genotype (allele_1/allele_2): ^6^ 03:02/99:99, 03:02/02, 06:04/03:02, 03:01/02, 06:03/03:02, 02/99:99, 06:04/02, 06:04/99:99, 03:01/03:02. 06:04/03:04, and. ^7^ 06:02/03:02, 06:02/99:99, 06:02/02, 06:03/99:99, 03:01/99:99, 06:02/03:01, 06:03/03:01, 06:04/03:01, 06:03/02, 03:04/99:99, 03:04/02, 06:02/03:04, 99:99/99:99. Note bene: 99:99 = remaining alleles.

**Table 2 nutrients-11-01980-t002:** Relative change in mean neonatal levels of cytokines, adipokines and other proteins involved in the immune response with 95% confidence bands linked with every doubling in neonatal whole blood zinc content—results from models without correction for multiple testing.

Outcome	Variable	Univariate Model	*p*-Value	Multivariate Model	*p*-Value
IL-1β	WB-Zinc content	1.00 (0.89; 1.12)	1.00	0.97 (0.87; 1.09)	0.64
IL-4	WB-Zinc content	0.99 (0.92; 1.06)	0.73	0.97 (0.90; 1.04)	0.37
IL-6	WB-Zinc content	1.08 (0.99; 1.19)	0.10	1.07 (0.96; 1.19)	0.21
IL-8	WB-Zinc content	0.98 (0.92; 1.04)	0.49	1.02 (0.97; 1.07)	0.52
IL-10	WB-Zinc content	0.98 (0.83; 1.16)	0.81	1.01 (0.86; 1.19)	0.91
IL-12	WB-Zinc content	1.08 (0.98; 1.20)	0.12	1.02 (0.93; 1.12)	0.61
IFN-γ	WB-Zinc content	1.02 (0.93; 1.11)	0.71	1.02 (0.92; 1.12)	0.73
TNF-α	WB-Zinc content	1.07 (0.98; 1.16)	0.15	1.01 (0.93; 1.10)	0.85
TGF-β	WB-Zinc content	1.06 (0.97; 1.14)	0.20	1.01 (0.93; 1.09)	0.83
Adiponectin	WB-Zinc content	1.02 (0.96; 1.10)	0.50	0.99 (0.93; 1.06)	0.86
Leptin	WB-Zinc content	0.97 (0.90; 1.05)	0.48	0.99 (0.93; 1.07)	0.90
CRP	WB-Zinc content	0.91 (0.82; 1.01)	0.08	0.97 (0.87; 1.08)	0.55
MBL	WB-Zinc content	1.04 (0.93; 1.17)	0.44	1.05 (0.93; 1.18)	0.41
sTREM-1	WB-Zinc content	0.99 (0.89; 1.11)	0.90	0.95 (0.85; 1.05)	0.31

Bold letters indicate significance at the two-sided 5% level. Covariates included in the multivariate models are neonatal whole blood Zinc (WB-Zinc) content, case status (cases are children who developed childhood type 1 diabetes (T1D), and controls who did not develop T1D during childhood), birthweight and period. IL, interleukin; IFN-γ, interferon gamma; TNF-α, tumor necrosis factor alpha; TGF-β, transforming growth factor beta; CRP, c-reactive protein; MBL, mannose-binding lectin; sTREM-1, soluble triggering receptor expressed on myeloid cells.

**Table 3 nutrients-11-01980-t003:** Relative change in mean neonatal levels of cytokines, adipokines and other proteins involved in the immune response with a 95% confidence bands linked with every doubling in neonatal whole blood zinc content—results from univariate models with stratification by case status and no correction for multiple testing.

Outcome	Variable	Univariate Model (Cases)	*p*-Value	Univariate Model (Controls)	*p*-Value
IL-1β	WB-Zinc content	0.97 (0.83; 1.14)	0.74	1.03 (0.89; 1.20)	0.66
IL-4	WB-Zinc content	0.99 (0.90; 1.09)	0.86	0.98 (0.90; 1.08)	0.71
IL-6	WB-Zinc content	1.12 (0.99; 1.26)	0.84	1.05 (0.91; 1.22)	0.50
IL-8	WB-Zinc content	0.99 (0.91; 1.08)	0.80	0.97 (0.90; 1.05)	0.43
IL-10	WB-Zinc content	0.97 (0.76; 1.22)	0.78	0.99 (0.78; 1.27)	0.96
IL-12	WB-Zinc content	1.07 (0.95; 1.20)	0.25	1.10 (0.96; 1.26)	0.17
IFN-γ	WB-Zinc content	1.04 (0.92; 1.16)	0.54	0.99 (0.87; 1.14)	0.94
TNF-α	WB-Zinc content	1.03 (0.91; 1.17)	0.61	1.11 (0.99; 1.24)	0.08
TGF-β	WB-Zinc content	1.04 (0.95; 1.13)	0.40	1.07 (0.95; 1.21)	0.24
Adiponectin	WB-Zinc content	1.00 (0.94; 1.07)	0.95	1.05 (0.94; 1.18)	0.38
Leptin	WB-Zinc content	0.98 (0.88; 1.08)	0.66	0.98 (0.89; 1.06)	0.57
CRP	WB-Zinc content	0.93 (0.81; 1.07)	0.32	0.88 (0.76; 1.04)	0.13
MBL	WB-Zinc content	1.05 (0.89; 1.24)	0.59	1.04 (0.88; 1.24)	0.65
sTREM-1	WB-Zinc content	0.95 (0.82; 1.10)	0.49	1.05 (0.91; 1.20)	0.51

Case status (cases are children who developed childhood type 1 diabetes (T1D) and controls who did not develop T1D during childhood). IL, interleukin; IFN-γ, interferon gamma; TNF-α, tumor necrosis factor alpha; TGF-β, transforming growth factor beta; CRP, c-reactive protein; MBL, mannose-binding lectin; sTREM-1, soluble triggering receptor expressed on myeloid cells-1.

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
