# Peer review of "Perinatal Whole Blood Zinc Status and Cytokines, Adipokines, and Other Immune Response Proteins"

_nutrients, 2019, doi:10.3390/nu11091980_

Round 1

Reviewer 1 Report

Dear Authors

Your article is very interesting and well documented

I have only one remark and one question :

the remark : page 2 line 87 : the significance of T1D is not explained ( probably type 1 diabetes) 

the question : why did you take as cases patients with type 1 diabetes ? and not another disease like pneumonia for example and so on ?

Best regards

Author Response

Reply to reviewers

We are grateful for the positive and constructive comments, and the opportunity to revise our manuscript. Below, we respond to each of the reviewers' comments.

Reviewer: 1

Comment 1.1:

Your article is very interesting and well documented

Reply 1.1:

Thank you very much for your kind comment.

Comment 1.2:

Ihave only one remark and one question: 

the remark: page 2 line 87: the significance of T1D is not explained ( probably type 1 diabetes)  

Reply 1.2:

Thank you for your comment. We have included this piece of text regarding the diagnosis criteria:

”Individuals were randomly selected from an original case-control study conducted by our research group aiming at examining predictors of childhood and adolescent type 1 diabetes (T1D) [21]. The focus of the current study is the association between zinc and a large number of cytokines, adipokines and other immune response proteins in healthy/non-infected newborns. Cases were those who later developed T1D, and controls were those who did not. Briefly, T1D diagnoses were retrieved from the Danish National Patient Registry based on ICD codes (ICD-8 249 and 250. ICD-10 DE 10.x–14.x) and were cross-validated using the Danish Childhood Diabetes Registry (DanDiabKids) to secure a T1D diagnosis [22]. DanDiabKids, established in 1996, is a nationwide register collecting data from all Danish paediatric diabetes clinics treating patients with T1D aged 0–18 years.” 

Comment 1.3:

the question: why did you take as cases patients with type 1 diabetes ? and not another disease like pneumonia for example and so on ? 

Reply 1.3:

A reasonable question. We only had data (immune markers and zinc measurements) on healthy neonates that later developed T1D and those who did not (controls). But our main question was also concerning the association between whole blood zinc levels and an array of immune markers in healthy neonates.

Reviewer 2 Report

The authors present their statistical analysis of observations regarding zinc and a variety of cytokines, all made from banked newborn, dried blood spots in the Danish Newborn Screening Biobank. The point of the paper is that in this cohort, from 1990s Danish, uninfected newborns, there were no correlations between zinc levels and detection of each of the major immune system signals.  The authors contextualize this to other, similar observations with different demographic characteristics (the authors suggest that the subjects of this work are all ethnically "Scandinavian" but do not state this definitively).

These analyzed data are of potential merit for publication given the very different results seen among published studies on this topic. 

However, there are some major methodological points that should be addressed before such a manuscript enters the published record:

With no prior background or subsequent clear/obvious analysis and justification the authors note that half of the samples were babies ultimately diagnosed with type I diabetes.  This seems to have been done intentionally, but none of the presented analyses take this obviously into account.  While the authors note that "case-status" is included with the multivariate model, the lack of transparency of what this model actually entails mitigates the merit of the presentation.

The authors do not describe their method(s) for the random draws of blood spot samples.

I recommend as a best practice that all raw data, code, model structures, and workflows be deposited with the paper as an open resource, or directly linked to another widely accepted repository.

Real values are never presented for any of the target analytes.  These are necessary (point 3 above) and should be compared to reference, expected ranges.

Furthermore, there are numerous minor issues, that accumulated lead the reader to question the validity of the work.  These issues give the impression of something that was put together in a very rushed manner and question whether or not the work was actually hypothesis driven (rather than just mined out of existing data from another paper by the same group):

The authors specifically state the method used for zinc analysis early on, but don't mention how they detected cytokines until deep into the methods of the paper.

Most of the paper contains glaring typographical errors; these are too numerous to list. Some are obviously the product of auto-correct software.  The authors should judiciously edit the copy of the current work before re-submission.

The authors rely heavily on two reviews for the immune portion of the introduction (though, alternatively, a decent job is done with contextualization of zinc population literature in the discussion). Furthermore, the sample presented herein is suspiciously similar to another paper that this group already published (REF 20).

The listing/grouping of subject proteins is odd in the abstract.  For example, "iiii." appears in the abstract, but not later in the paper; abbreviations are not consistent (e.g. IFN-λ and TNF-α).

One of the stated strengths is a "large sample size".  This is subjective, and I would argue the sample size is small to moderate given the type and availability of samples, and a clear published record by the authors of capability with the techniques (again, suspiciously similar to another paper).  While this work is larger than many of the studies cited in the discussion, these are also exceedingly small.

A more specific listing/description of the Luminex assay performed is necessary within the paper, rather than relying on a chain of referenced papers assuming the same technique.

Author Response

Reply to reviewers

We are grateful for the positive and constructive comments, and the opportunity to revise our manuscript. Below, we respond to each of the reviewers' comments.

Reviewer: 2

Comment 2.1:

The authors present their statistical analysis of observations regarding zinc and a variety of cytokines, all made from banked newborn, dried blood spots in the Danish Newborn Screening Biobank. The point of the paper is that in this cohort, from 1990s Danish, uninfected newborns, there were no correlations between zinc levels and detection of each of the major immune system signals.  The authors contextualize this to other, similar observations with different demographic characteristics (the authors suggest that the subjects of this work are all ethnically "Scandinavian" but do not state this definitively).

These analyzed data are of potential merit for publication given the very different results seen among published studies on this topic. 

Reply 2.1:

Thank you very much for taking your time to thoroughly review our paper. Regarding your comment about ethnicity we have included the following text:

“The DNSB is majorly comprised of Danish ethnic children. In a study (n = 3,956), ~94% of the newborns had mothers of Danish origin, 1.4% had mothers from other western countries and the remaining newborns had mothers from non-western countries [24].”

Further, all children are born in Denmark as already stated under “Study design and Sampling”:

“Newborn dried blood spots (NDBS), from children born in Denmark between January 1991 and November 1998,…”

Comment 2.2:

With no prior background or subsequent clear/obvious analysis and justification the authors note that half of the samples were babies ultimately diagnosed with type I diabetes.  This seems to have been done intentionally, but none of the presented analyses take this obviously into account.  While the authors note that "case-status" is included with the multivariate model, the lack of transparency of what this model actually entails mitigates the merit of the presentation.

Reply 2.2:

We understand your point and have also thought of how to include this information, maybe we haven’t done a good enough job. We have included the following text in the paper:

”Individuals were randomly selected from an original case-control study conducted by our research group aiming at examining predictors of childhood and adolescent type 1 diabetes (T1D) [21]. The focus of the current study is the association between zinc and a large number of cytokines, adipokines and other immune response proteins in healthy/non-infected newborns.Cases were those who later developed T1D, and controls were those who did not. Briefly, T1D diagnoses were retrieved from the Danish National Patient Registry based on ICD codes (ICD-8 249 and 250. ICD-10 DE 10.x–14.x) and were cross-validated using the Danish Childhood Diabetes Registry (DanDiabKids) to secure a T1D diagnosis [22]. DanDiabKids, established in 1996, is a nationwide register collecting data from all Danish paediatric diabetes clinics treating patients with T1D aged 0–18 years.” 

Of notice, previously published articles from our group only found very few and small differences between cases (T1D) and controls (no T1D) regarding the immune markers (PMID: 27871914) and no differences in zinc levels (PMID: 27873432), but that does not exclude potential effect modification (interaction) of case-status on zinc ~ immune marker effects.

Therefore, we have run analyses stratifying on case-status to ensure the reader of no sign of interaction of case-status on main effects. We lose power, but results remain the same—the results are included as a new Table 3, and the result from the previous Table 3 are now shown in Table 3.  

We have included the following changes in the manuscript:

Statistics:

“Neonatal WB-Zinc content was log2-transformed, due to a log-normal distribution. Results are therefore interpreted as relative change (RC) in mean levels of immune markers for each doubling in neonatal WB-Zinc content. We analyzed the data overall and stratified by case-status.”

Results:

“3.3. Unadjusted Models with Statification on Case-Status

We found no signs of effect modification of case-status (neonates that later developed T1D versus neonates that did not) on our main effects. Results are presented in Table 3.“

In accordance with yourcomments we have includedthestratified analyses in Table 3, and thepreviousoriginal Table 3 (analyses with correction for multiple testing) has now been changed to Supplementary table 1.

Comment 2.3:

The authors do not describe their method(s) for the random draws of blood spot samples.

Reply 2.3:

To address your comment, we have included the following text in the statistics section:

“The random sampling of 400 individuals (200 cases and 200 controls) from the larger cohort of 2086 cases and 4172 controls [21]was done by a random number generator in Statistical Analysis Software (SAS) version 9.3 using the floor-statement and was done on the cases born from 1991-1998 (as the same filter paper was used in this period). Each case was originally matched with 2 controls on date-of-birth, but for this nested study we only included the first control found for each case.”

Comment 2.4:

 recommend as a best practice that all raw data, code, model structures, and workflows be deposited with the paper as an open resource, or directly linked to another widely accepted repository.

Reply 2.4:

If editor wishes, we are happy to deposit/publish the statistical code. We are prohibited from depositing raw data due to current legislation in the European Union (the General Data Protection Rule), lack of consent from participants to publicly distribute personal data, and strict rules by the national registers not to reveal any personal identifiable information on participants.   

Comment 2.5:

Real values are never presented for any of the target analytes.  These are necessary (point 3 above) and should be compared to reference, expected ranges.

Reply 2.5:

Absolute values for the biomarkers have been published previously for the same sample (PMID:

30836628). We have referred to these values in the paper in p 4 line 164-165:

“Absolute levels of the cytokines, adipokines and other proteins involved in the immune response stratified by case-status are presented elsewhere [31].”

Comment 2.6:

Furthermore, there are numerous minor issues, that accumulated lead the reader to question the validity of the work.  These issues give the impression of something that was put together in a very rushed manner and question whether or not the work was actually hypothesis driven (rather than just mined out of existing data from another paper by the same group):

The authors specifically state the method used for zinc analysis early on, but don't mention how they detected cytokines until deep into the methods of the paper.

Reply 2.6:

We are not completely sure of which minor issues you are addressing, but we take our research serious and it would be unpleasing if readers questioned the validity of our work.  As we indirectly state in the end of the introduction, our hypothesis was the following: “Neonatal whole blood zinc content is associated with systemically circulating immune markers”. If one looks through the literature little is known regarding this topic in early life, but zinc has in many study designs been linked to immunological health. Further, zinc and iron (as you mention our previous paper) are the most abundant trace elements in the human body and deficiencies in these two trace elements are among the most common nutritional deficiencies worldwide. To conclude, we have not just mined out this paper from existing data, but rather seen a potential in such unique data. We have used a fair amount of research resources on this paper and think it brings novelty into this field of study.

Regarding mentioning how cytokines were measured we have moved this paragraph just under “Exposure Assessment” in the Methods section. 

Comment 2.7:

Most of the paper contains glaring typographical errors; these are too numerous to list. Some are obviously the product of auto-correct software.  The authors should judiciously edit the copy of the current work before re-submission.

Reply 2.7:

We naturally regret these errors and have now corrected them. As non-native English speakers, we appreciate your comments on this.  

Comment 2.8:

The authors rely heavily on two reviews for the immune portion of the introduction (though, alternatively, a decent job is done with contextualization of zinc population literature in the discussion). Furthermore, the sample presented herein is suspiciously similar to another paper that this group already published (REF 20).

Reply 2.8:

 We appreciate that you have taken your time to look up the article by Kyvsgaard et al (PMID:  

27873432), which is referenced in this manuscript. In that article, we did not investigate the association of zinc with immune biomarkers.Thus, the overlap is only by study-population and some basic methodology, which is no different from other studies, hypothesizing different  questions on the same cohort.

Comment  2.9:

The listing/grouping of subject proteins is odd in the abstract.  For example, "iiii." appears in the abstract, but not later in the paper; abbreviations are not consistent (e.g. IFN-λand TNF-α).

Reply 2.9:

The grouping of the cytokines and the adipokines follows the NIH indexed MESH headings. The third group was designated by us as “other immune response proteins” as this group consisted of various other proteins classified elsewhere. The grouping is also consistent with the title. We adjusted the abstract such that the groups i, ii, and iii follow the same order as in the main text.

Comment 2.10:

One of the stated strengths is a "large sample size".  This is subjective, and I would argue the sample size is small to moderate given the type and availability of samples, and a clear published record by the authors of capability with the techniques (again, suspiciously similar to another paper).  While this work is larger than many of the studies cited in the discussion, these are also exceedingly small

Reply 2.10:

You are right this is relative, so we revised the limitations section:

“This study is larger than many of the studies previously published on the association of early life zinc exposure and immune biomarkers [9,14,16,17,35,39].”

Comment 2.11:

A more specific listing/description of the Luminex assay performed is necessary within the paper, rather than relying on a chain of referenced papers assuming the same technique.

Reply 2.11:

We have followed your recommendations regarding including more about the multiplexed sandwich immunoassay assay. See the revised section below:

“The NDBS were analyzed using a multiplexed sandwich immunoassay, based on flowmetric Luminex xMAP®technology, which can measure up to 25 inflammatory markers simultaneously on NDBS [29]. Thefollowing analytes were quantified: i. cytokines; IL-1β, IL-4, IL-6, IL-8, IL-10, IL-12(p70), IFN-γ, TNF-α, transforming growth factor beta (TGF-β); ii. adipokines; leptin and adiponectin; and iii. CRP and mannose-binding lectin (MBL) involved in innate immunity, and soluble triggering receptor expressed on myeloid cells1 (sTREM-1) belonging to the immunoglobin superfamily.

In all assays, matched pairs were run together to avoid batch effects/ interassay variation. Biomarker analyses are described in detail elsewhere [29]. Quality control of the analysis were made using mouse IL-6 as an internal analyte added to the extraction buffer to detect pipetting errors, and biotinylated beads to detect signal errors (a more detailed description are found in Skogstrand [30]). Calibration curves were used on each plate together with one high and two low controls. Samples, calibrators, and controls were analyzed in duplicates.”

Round 2

Reviewer 2 Report

The authors have improved their paper substantially and in good faith.  The editor should be sure to address the comment on the European Union General Data Protection Rule.

Author Response

Dear reviewer 

We thank you again for taking your time to review our paper. 

Comment 1.1:

The authors have improved their paper substantially and in good faith.  The editor should be sure to address the comment on the European Union General Data Protection Rule.

Reply 1.1:

Thank you